# Extending the Frontiers of Electronic Commerce Knowledge through Cybersecurity

Daniela Roxana Vuță *[ID], Eliza Nichifor [ID], Ovidiu Mircea Țierean, Alexandra Zamfirache [ID], Ioana Bianca Chițu [ID], Tiberiu Foris and Gabriel Brătucu [ID]

Faculty of Economic Sciences and Business Administration, Transilvania University of Brașov, 500036 Brașov, Romania; eliza.nichifor@unitbv.ro (E.N.); ovidiu.tierean@unitbv.ro (O.M.Ț.); alexandra.zamfirache@unitbv.ro (A.Z.); ioana.chitu@unitbv.ro (I.B.C.); tiberiu.foris@unitbv.ro (T.F.); gabriel.bratucu@unitbv.ro (G.B.)
* Correspondence: daniela.vuta@unitbv.ro

**Abstract:** As technology becomes more sophisticated so do cyber-attacks. The resilience of electronic commerce organisations represents a critical point nowadays because it influences consumer and digital business behaviour. In this context, the cybersecurity and electronic commerce knowledge were reviewed as a unit. The main aim of this paper is to support researchers and managers in understanding the theoretical framework and to provide a knowledge-based model. To achieve this aim, the authors performed an analysis of 14,585 papers from the Web of Science Core Collection that generated two visualized networks, analyzed with the metrics mean silhouette, modularity, betweenness centrality, and citation bursts in the context of digital resilience. The mapping process results show that the human factor represents the central junction with the fear of cyber-attacks and the perception of online shopping as risky. The adoption of electronic commerce and mobile commerce are two challenging research lines in the global economic resilience. Their adoption enabled by big data, artificial intelligence, machine learning, and even blockchain technology can strengthen resilience even when cybersecurity education is needed.

**Keywords:** electronic commerce; cybersecurity; digital resilience

## 1. Introduction

The entire world faced an unprecedented situation in 2020, the COVID-19 pandemic [1], which led to another major series of changes. For example, in Singapore, cybercrime accounts for almost half of the total crimes reported, and the number of ransomware attacks rose by 154% [2]. A recent study [3] demonstrated that European countries develop different sensitivities to cybersecurity issues. The Craigen et al. study revealed the definition of cybersecurity as "the organization and collection of resources, processes, and structures used to protect cyberspace and cyberspace-enabled systems from occurrences that misalign the jure from de facto property rights" [4]. In addition, this topic is the subject of national interest because the United States consider cyber offence and cyber defence as "vital elements of security strategies", these being more related to national policies and defence strategies [5]. Digitization and cybersecurity play a key role in the NGEU (Next Generation EU), which are two of the main drivers of sustainable development [6]. The purpose of the NGEU is to support the sustainability-resilience relationship and the importance of cybersecurity is highlighted in this way. It becomes necessary to protect the information and data of citizens and companies (this paper highlights the need for new studies in the context of electronic commerce [3]. All of this generates a positive relationship between information systems, sustainability, and resilience [3].

All of these challenges have decisively influenced various areas, especially electronic commerce and digital business—both company and consumer behaviour.

Contrastingly, the study of Dirgantar et al. [7] brings forward the importance of the information system for electronic commerce organisations, whereby their results confirmed that system characteristics such as quality, information quality, or service quality, influence the level of use and user satisfaction of customers. An information system refers to the process of collecting, processing, storing, and transmitting relevant information to assist decision-making and foster operations in any type of organization [8]. It involves a set of interconnected components such as hardware, software devices, human resources, and networks that contribute to improving organizational productivity, efficiency, and effectiveness while ensuring business survival and resilience [9,10]. In the absence of a resilience mechanism to overcome interruptions in business, as usual, activities in electronic commerce could no longer function [11]. Resilience may be analyzed from the perspective of cyber threats to the information system, as cyber resilience can help organizations to anticipate and manage them [12,13]. At the same time, increasing resilience in electronic commerce can be achieved using Artificial Intelligence, which can contribute to 'readying supply chains to reduce their risk of disruption' [14]. To combat threats, function accordingly, and take on challenges, information systems must adapt through structural and operational resilience techniques [15]. So, to prevent companies from losing important components of their information systems, cybersecurity is the key to success [16,17].

Contrastingly, the COVID-19 pandemic forced 'late' consumers to reduce the digital gap, directing them to electronic commerce platforms [18], but it also demonstrated the resilience of consumers increases [19–21]. Digital resilience is not a new concept, its definition has existed in literature for many years [22]. It is "the ability for an organization to rapidly adapt to business disruptions by leveraging digital capabilities to not only restore business operations but also capitalize on the changed conditions" [23]. Contrastingly, cyber resilience is the capacity to protect data and electronic systems from cyber-attacks, but also, to rapidly resume business operations in the case of a successful attack event. [24]. Knowing these aspects, the digital resilience of electronic commerce was ascertained [25], as was the importance of long-term planning, and the need for strategic skills to make the right decisions for traders [26]. Electronic commerce is a vital element in worldwide economic resilience [27] and the adoption of digitisation and digital platforms is an 'escape hatch' in the development strategy and resilience of organisations [28–30]. Even though there are still potential consumers who perceive online shopping as risky [30], it is concerned about both the resilience of the retailer's information system and the security of personal data [31,32].

All of these facts have revolutionized people's lives and influenced users' buying habits, and the way that companies sell their products or services [33]. As such, information system security became a major challenge for modern organisations, which has led to a growing interest from academia and researchers in information security knowledge. [34].

To better understand the context regarding electronic commerce, cybersecurity, and digital resilience, the authors illustrated Figure 1.

In this context, the authors tried to find resilient opportunities for electronic commerce, by aiming to analyse the research knowledge fields, respectively. The study was performed with the method of visual network analysis with Java technology by constructing and observing the connections between the scientific resources.

The results were conducted to relevant outcomes for proposing a new conceptual model by including all relevant key concepts regarding the knowledge fields included in the study. (The Resilience Right Decisions for Electronic Commerce and Cybersecurity Model).

All the concepts existing in the literature, the methods used, and the results are presented in the subsequent sections. The article finishes with a discussion and conclusion section, followed by two sections that highlight the importance of the study and the its necessity to the scientific community.

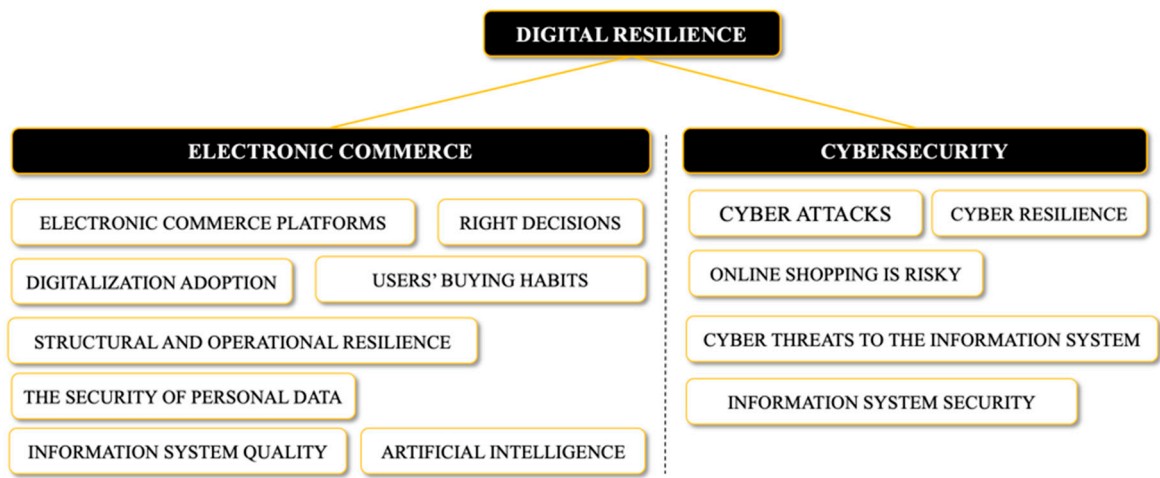

**Figure 1.** Literature review summary designed by authors.

## 2. Materials and Methods

Due to the complexity of the materials and method used, this section is presented in three parts. The first one is allocated to data collection, followed by project configurations, whereas the final part introduces details about the analysis method. For a better understanding, Table 1 introduces the abbreviations used before any methodological considerations.

**Table 1.** List of abbreviations.

| Abbreviation | Meaning |
| --- | --- |
| Q | Modularity metric |
| LLR | Log-likelihood ratio |
| S | Mean silhouette |

### 2.1. Data Collection

Following the framework guidelines published by Mukherjee et al., 2022 [35] and Donthu et al., 2021 [36], the authors created a comparative narrative report based on three areas of knowledge, respectively. The primary source for the dataset was the Web of Science. Using the basic search of the database, bibliometrics such as topic or keyword were applied to the timespan between 2000 and 2022. The analysis started by inserting the keyword in the search field for the following terms related to the topics of interest: electronic commerce, cybersecurity, and digital resilience.

This action was performed first because the authors were interested in mapping the network and visualizing the connections between all research publications related to these fields. The basic search found only two results (papers), which was an astonishing surprise as the authors discovered the potential of the research knowledge in these areas. Therefore, the study framework was supposed to analyse each field one-by-one, by creating individual databases with the scientific papers published. Firstly, "electronic commerce" was searched for, using the period between 2000 and 2022. The search generated 1579 results from the Web of Science Core Collection. They were displayed by relevance, aiming to increase the quality of papers included in the study. Moreover, the authors observed and collected the number of citations of the most relevant published paper in the field. The same process was followed for each topic, resulting in a generous number of papers as follows in Table 2. Each of them were downloaded as plain text files and prepared in a data file for analysis.

**Table 2.** Basic search results for topics of interest in the timespan between 2000–2022.

| Topic | Number of Results in Web of Science Core Collection | Citations of the Most Relevant Paper |
|---|---|---|
| Ecommerce | 1570 | 10 |
| Cybersecurity | 10,388 | 8 |
| Digital resilience | 2627 | 14 |

### 2.2. Project Configuration

The network construction process was represented by configuring the time-slicing parameter from 2000 to 2022, and text processing by checking title, abstract, author keywords (DE), and keywords. After that, the network was set up with the default node type (cited reference). The selection criteria for all the items included in the study are represented by top N = 50, which means that the software selected the fifty most cited items for each slice to construct the network. In the same row, the top N% per slice was set up to 10%, indicating that the selection was made for 10% of the most cited items or most frequent items per slice. The selection criteria used for each slice are the modified g-index (the alternative for the older h-index) and the scale factor $k = 25$.

$$g^2 = \leq k\Sigma_{\, i \leq g}c_i, \ k \in \mathbf{Z}^+ \tag{1}$$

where $k$ is the number of publications and $c$ is the number of citations.

### 2.3. Analysis Method

Several CiteSpace [37] features were used. The first one is a clustering function used to identify a specific theme, topic, or line of research. The second one is the metric modularity Q, which measures the extent to which a network can be discomposed into multiple components. Then, the mean silhouette highlights the overall structure of the network.

For example, the authors used this to show the homogeneity of the cluster. Its value ranges between −1 and 1, hence the higher the score, the more consistent the cluster members are. The labelling of the clusters process was performed to characterise the nature of identified clusters. The authors extracted the noun phrases from titles, keywords, and abstracts.

To find out where one may find the major areas of research, the authors set up the nodes and the font sizes, and the transparency of the links. After that, to see how these major areas are connected, the authors used the betweenness centrality indicator to highlight the nodes with high scores.

The burst detection process is supposed to use citation on burst indicator to find where the most active areas are in the field. The parameter provides information about the publications that attracted an extraordinary degree of attention during the time based on Kleinberg's algorithm [38]. To run the analysis, the selected node type was the keyword for a time span between 2000 and 2022, slice length = 1, and Top N = 100. Additionally, the citation burst history helped the authors to summarise the list of articles associated with citation bursts for each field.

Lastly, a cluster exploration was run to analyse the clusters in more depth and a timeline view was set up to position each cluster arranged on a horizontal timeline.

## 3. Results

The network mapping process generated the comparative outcome illustrated in Figure 2.

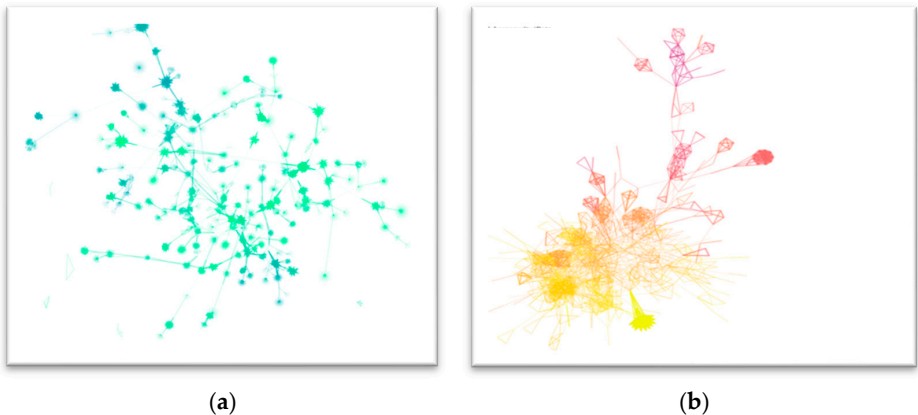

(**a**)                                        (**b**)

**Figure 2.** Visualized comparative analysis: (**a**) Network for electronic commerce field of research; (**b**) Network for cybersecurity field of research, designed by authors. Note: The more obvious the color, the more recent the publications.

*3.1. Electronic Commerce*

Running the project for this knowledge field, 1570 references were included in the study and due to software criteria, in the final, 1559 were qualified records (2000–2022). To visualise some relevant groupings for this research area, the clustering process generated 280 groups with the top research lines labelled according to the dimension of each one. The authors characterised the nature of the groupings by choosing to select noun phrases from the titles and showing labels by the log-likelihood ratio (LLR). The cluster labels are shown as follows in Figure 2. The modularity Q (0.9559) and the mean Silhouette S (0.9806), the metrics that describe the structural properties of the network, show a high homogeneity of the clusters and dense connections between nodes within them. Therefore, the network is divided into five co-citations clusters (Figure 3) and the most relevant ones with their sizes and top terms are presented as follows in Table 3.

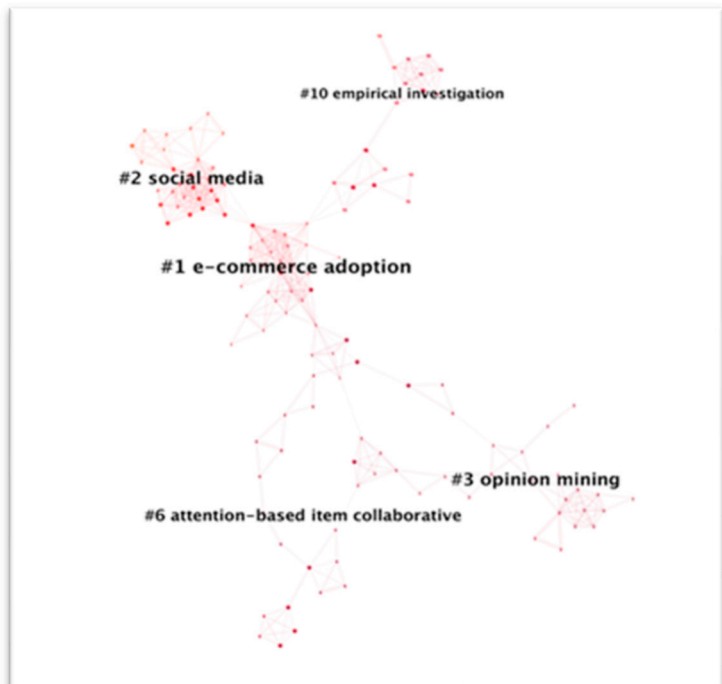

**Figure 3.** Labelled clusters for electronic commerce, designed by authors. Note: The more obvious the color, the more recent the publications.

**Table 3.** Top terms by cluster labels and indicators for electronic commerce.

| Cluster ID | Size | Silhouette | Mean (Year) | Label (LLR) |
|---|---|---|---|---|
| 1 | 31 | 0.962 | 2016 | **e-commerce adoption** (18.1, $1.0 \times 10^{-4}$); **influencing beliefs formation** (15.79, $1.0 \times 10^{-4}$); **social commerce** (15.79, $1.0 \times 10^{-4}$); SME travel agencies (15.79, $1.0 \times 10^{-4}$); **mobile commerce adoption** (13.49, 0.001) |
| 2 | 28 | 1 | 2012 | social media (23.99, $1.0 \times 10^{-4}$); latent transition analysis (23.99, $1.0 \times 10^{-4}$); trip experience (23.99, $1.0 \times 10^{-4}$); tourism design (19.05, $1.0 \times 10^{-4}$); smart tourism development (14.18, 0.001) |
| 3 | 27 | 0.985 | 2018 | opinion mining (21.68, $1.0 \times 10^{-4}$); fuzzy logic (21.68, $1.0 \times 10^{-4}$); salient research topics (17.97, $1.0 \times 10^{-4}$); **analysing e-wom** (14.31, 0.001); stochastic dominance (14.31, 0.001) |
| 6 | 22 | 0.978 | 2017 | attention-based item collaborative (21.02, $1.0 \times 10^{-4}$); **fast shipping ecommerce** (18.33, $1.0 \times 10^{-4}$); case study (18.33, $1.0 \times 10^{-4}$); inbound logistics operation (18.33, $1.0 \times 10^{-4}$); **purchasing attitude** (15.65, $1.0 \times 10^{-4}$) |
| 10 | 18 | 0.978 | 2015 | empirical investigation (22.2, $1.0 \times 10^{-4}$); big data perspective (22.2, $1.0 \times 10^{-4}$); online review helpfulness (22.2, $1.0 \times 10^{-4}$); specific word entropy (17.64, $1.0 \times 10^{-4}$); **purchasing behaviour** (13.14, 0.001) |

To find what the most active areas in the electronic commerce field of knowledge are, a burst detection was performed. To do so, the reference node was selected to showcase the landmark's papers in this field by this indicator [39].

Another useful result is related to the centrality indicator that shows how the major areas are connected, presenting the paper with the highest centrality papers [40], Bastl M, 2012, INT J OPER PROD MAN, 32, 650, Bennett D, 2012, INT J OPER PROD MAN, 32, 1281, Cai SH, 2010, J OPER MANAG, 28, 257.

Therefore, the burst detection by keywords determined the authors to present the big picture of the topics of interest in the electronic commerce field (Table 4).

**Table 4.** Burst detection by keywords for electronic commerce.

| Keywords | Strengths | Begin | End | 2000–2022 |
|---|---|---|---|---|
| social media | 5.89 | 2016 | 2019 | |
| online review | 5.82 | 2020 | 2022 | |
| destination marketing | 5.59 | 2007 | 2011 | |
| web service | 4.93 | 2001 | 2005 | |
| e-commerce | 4.65 | 2011 | 2013 | |
| social network | 4.64 | 2015 | 2017 | |
| sentiment analysis | 4.58 | 2018 | 2022 | |
| information search | 4.34 | 2009 | 2014 | |
| service | 3.98 | 2019 | 2022 | |
| web | 3.97 | 2011 | 2017 | |
| perceived risk | 3.93 | 2018 | 2019 | |
| data mining | 3.72 | 2012 | 2015 | |
| purchase intention | 3.61 | 2019 | 2022 | |
| online shopping | 3.49 | 2014 | 2019 | |
| experience | 3.34 | 2017 | 2019 | |
| electronic commerce | 3.28 | 2008 | 2013 | |
| tourism | 3.18 | 2008 | 2017 | |

In generating the top keywords report, the authors wanted to discover more details relating to it, so a mapping of clusters (Figure 4) by keyword node was performed. The results exhibit 146 clusters for the same time span; the selection criteria are represented by the Top 10% per slice and 1.00% nodes labelled, without pruning (network, N = 18,325, E = 57,424, and density = 0.0003).

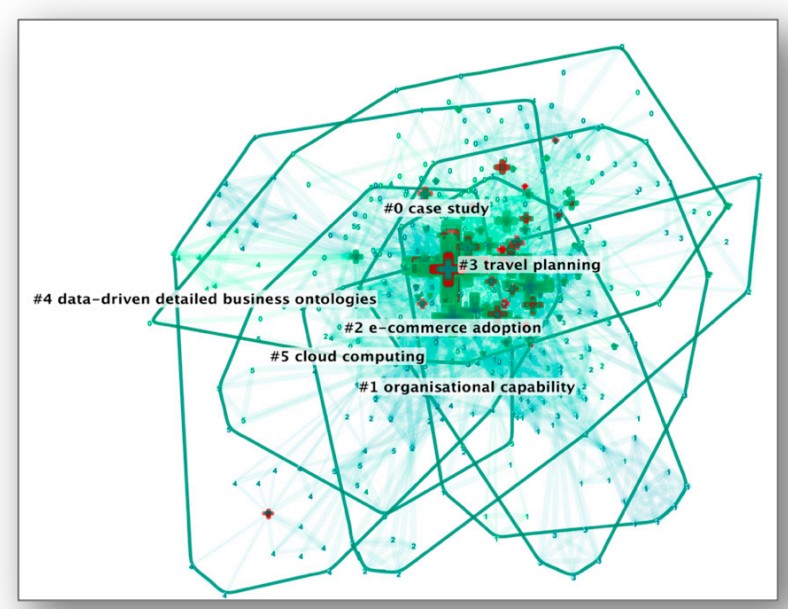

**Figure 4.** Cluster mapping by keyword node was performed (electronic commerce).

By title node, the electronic commerce research area is divided into thirteen co-citations clusters for 1559 qualified records. The timeline view illustrates the evolution of the main topics of research as follows in Figure 5. The generated data shows the most relevant topics in electronic commerce by clusters (Table 5).

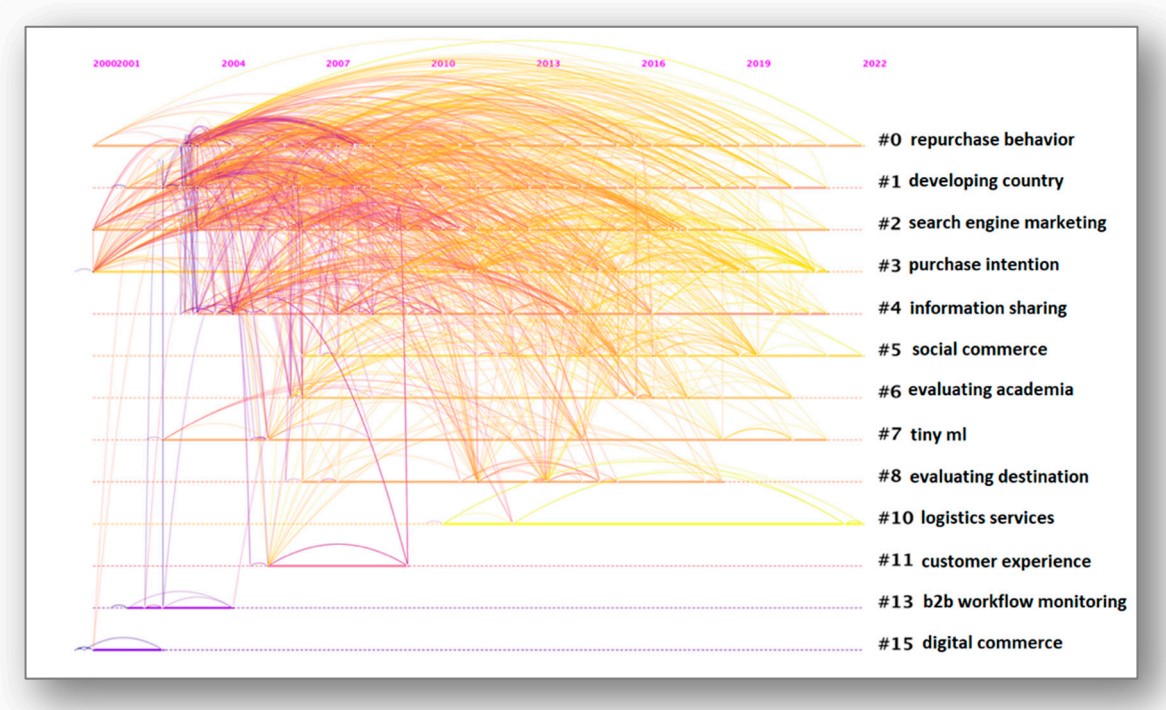

**Figure 5.** Timeline view for cluster labelling by title node.

**Table 5.** Most relevant topics in electronic commerce by clusters.

| Cluster ID | Size | Silhouette | Label (LLR) | Year | The Most Relevant Topics |
|:---:|:---:|:---:|:---:|:---:|:---|
| 0 | 42 | 0.643 | repurchase behaviour (144.12, $1.0 \times 10^{-4}$) | 2011 | factors and performance impact of electronic business [41] |
| 1 | 42 | 0.727 | a developing country (327.66, $1.0 \times 10^{-4}$) | 2010 | Electronic commerce adoption willingness and behaviour [42] |
| 2 | 38 | 0.736 | search engine marketing (255.83, $1.0 \times 10^{-4}$) | 2012 | search engine use [43] |
| 3 | 31 | 0.586 | purchase intention (194.09, $1.0 \times 10^{-4}$) | 2015 | Electronic commerce satisfaction [44] |

### 3.2. Cybersecurity

The labelling process of the clusters for cybersecurity was based on 10,245 qualified records and generated 234 groups with high homogeneity (S = 0.9268). Table 6 shows that the size of each cluster is relevant enough to present the major research lines.

**Table 6.** Clusters of cybersecurity are labelled by major research lines.

| Cluster ID | Size | Silhouette | Mean (Year) | Label (LLR) |
|:---:|:---:|:---:|:---:|:---|
| 0 | 99 | 0.92 | 2016 | **human factor** (1038.26, $1.0 \times 10^{-4}$); **machine learning** (760.94, $1.0 \times 10^{-4}$); health care (678.93, $1.0 \times 10^{-4}$); scoping review (601.43, $1.0 \times 10^{-4}$); smart grid (593.5, $1.0 \times 10^{-4}$) |
| 1 | 95 | 0.915 | 2017 | network intrusion detection (1797.74, $1.0 \times 10^{-4}$); **using machine** (933.96, $1.0 \times 10^{-4}$); objective comparison (863.01, $1.0 \times 10^{-4}$); IoT network (829.85, $1.0 \times 10^{-4}$); intrusion detection system (790.24, $1.0 \times 10^{-4}$) |
| 2 | 78 | 0.873 | 2016 | **blockchain technology** (2099.46, $1.0 \times 10^{-4}$); smart cities (1690.93, $1.0 \times 10^{-4}$); **blockchain technologies** (928.62, $1.0 \times 10^{-4}$); IoT device (826.53, $1.0 \times 10^{-4}$); **using blockchain** (643.47, $1.0 \times 10^{-4}$) |
| 3 | 69 | 0.914 | 2015 | adversarial machine learning (1756.66, $1.0 \times 10^{-4}$); adversarial example (1053.93, $1.0 \times 10^{-4}$); **deep learning** (861.22, $1.0 \times 10^{-4}$); **machine learning** (854.89, $1.0 \times 10^{-4}$); adversarial attack (660.04, $1.0 \times 10^{-4}$) |
| 4 | 65 | 0.884 | 2016 | industrial control system (2556.12, $1.0 \times 10^{-4}$); in-vehicle network (1172, $1.0 \times 10^{-4}$); **attack detection** (894.82, $1.0 \times 10^{-4}$); case study (782.27, $1.0 \times 10^{-4}$); **behavioural model** (770.93, $1.0 \times 10^{-4}$) |
| 5 | 61 | 0.944 | 2014 | **national cybersecurity** (682.86, $1.0 \times 10^{-4}$); **shared responsibility** (610.77, $1.0 \times 10^{-4}$); **global cybersecurity** (538.74, $1.0 \times 10^{-4}$); political economy (472.75, $1.0 \times 10^{-4}$); theorising cyber coercion (466.76, $1.0 \times 10^{-4}$) |
| 6 | 54 | 0.949 | 2012 | load redistribution attack (1494.73, $1.0 \times 10^{-4}$); advanced metering infrastructure (747.5, $1.0 \times 10^{-4}$); power system adequacy assessment (741.01, $1.0 \times 10^{-4}$); power grid (656.86, $1.0 \times 10^{-4}$); 3d printing cybersecurity (656.86, $1.0 \times 10^{-4}$) |

To see how the major areas are interconnected (Figure 6), the betweenness centrality analysis was used. By the reference node, the item with the higher centrality (47) is associated with the process of attack detection using deep learning for IoT[] [45], which is ranked in Cluster #1.

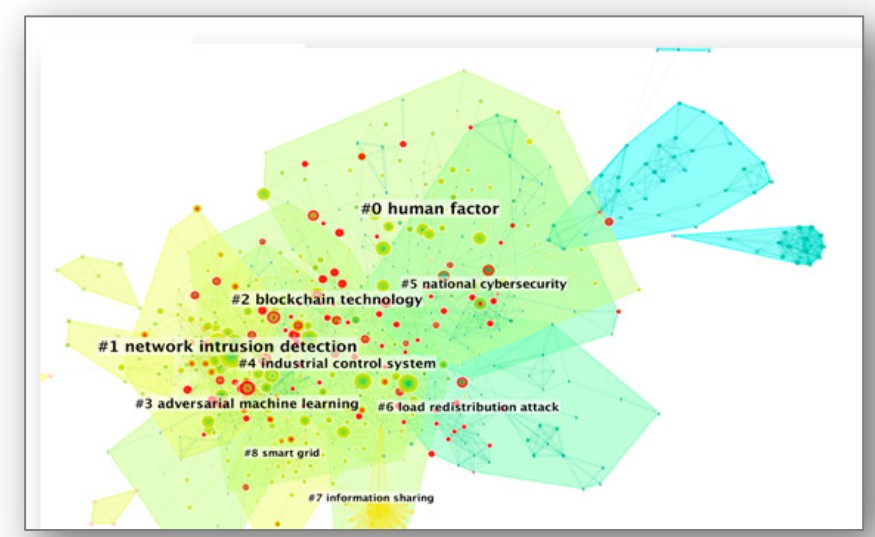

**Figure 6.** The interconnection of major lines for cybersecurity.

Afterwards, the authors discovered what the most active areas in cybersecurity are, with the help of burst detection for reference nodes. The results showed that three of the topics are related to the law applicable to Cyber operations [46] and cybersecurity in the Smart Grid (next-generation power systems) [47].

To enable a comparison between electronic commerce and cybersecurity fields of research by keyword node, a new analysis for 10,242 qualified records was run. The major themes for cybersecurity are divided into sixty-nine clusters with the three preeminent presented in Table 7.

**Table 7.** Labels and clusters for cybersecurity by keyword node.

| Cluster ID | Size | Silhouette | Mean (Year) | Label (LLR) |
|---|---|---|---|---|
| 0 | 142 | 0.698 | 2017 | cybersecurity awareness $(4192.79, 1.0 \times 10^{-4})$ |
| 1 | 132 | 0.628 | 2017 | smart cities $(5279.12, 1.0 \times 10^{-4})$ |
| 2 | 105 | 0.725 | 2016 | machine learning $(6747.1, 1.0 \times 10^{-4})$ |

The Silhouette means shows a relatively high homogeneity which is associated with the large sizes of the clusters. Therefore, the keyword nodes associated with the most relevant citers for each cluster highlight the impact of their work on the related topic of research. By comparing cybersecurity with electronic commerce (Figure 7), the authors aimed to find a possible nexus path between them. Analysing the models, cluster #0, cybersecurity awareness in cybersecurity, and cluster #2, electronic commerce adoption emerged.

Both seem to represent major areas in each network; therefore, a detailed analysis by keyword nodes was performed for further analysis of cybersecurity knowledge. The top keywords are shown in Table 8.

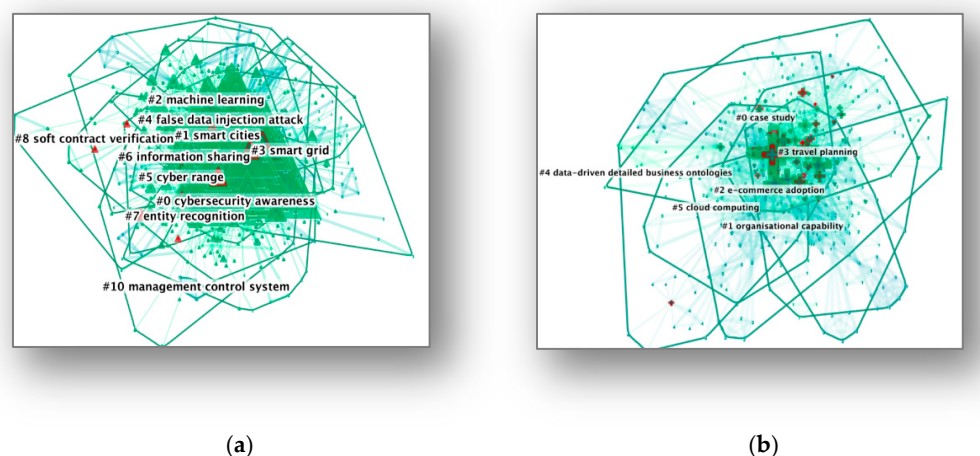

(**a**)                                                       (**b**)

**Figure 7.** Comparative analysis by cluster mapping (keyword node): (**a**) clusters for cybersecurity; (**b**) clusters for electronic commerce.

**Table 8.** Top keywords for cybersecurity research line.

| Keywords | Strengths | Begin | End | 2000–2022 |
|---|---|---|---|---|
| cyber security | 19.22 | 2012 | 2017 | |
| smart grid | 12.36 | 2010 | 2017 | |
| cybersecurity education | 7.76 | 2014 | 2018 | |
| vulnerability assessment | 6.28 | 2005 | 2018 | |
| information security | 6.21 | 2008 | 2014 | |
| moving target defence | 5.88 | 2016 | 2018 | |
| cloud computing | 5.82 | 2011 | 2017 | |
| cyber defence | 5.42 | 2014 | 2018 | |
| information sharing | 5.42 | 2015 | 2017 | |
| static analysis | 5.17 | 2018 | 2019 | |
| computer security | 5.09 | 2009 | 2015 | |
| critical infrastructure | 5 | 2015 | 2017 | |
| security | 4.99 | 2013 | 2015 | |
| crime | 4.94 | 2017 | 2018 | |
| attack graph | 4.93 | 2012 | 2018 | |
| data protection | 4.89 | 2017 | 2018 | |
| social network | 4.87 | 2016 | 2018 | |
| big data | 4.83 | 2012 | 2018 | |
| software security | 4.81 | 2016 | 2018 | |
| game theory | 4.63 | 2014 | 2018 | |
| situation awareness | 4.46 | 2016 | 2019 | |
| smart home | 4.42 | 2019 | 2020 | |
| critical infrastructure protection | 4.4 | 2015 | 2016 | |
| malware | 4.16 | 2019 | 2020 | |
| cybersecurity training | 4.13 | 2017 | 2018 | |

## 4. Discussion

The paper enriches the literature by bringing together three major themes that are strongly inter-connected. Remembering Figure 1 presented in the introduction (where the existing scientific contribution related to electronic commerce, cybersecurity, and digital resilience were illustrated), the authors exhibit a single big picture of the findings, connecting the current state of knowledge. Thus, to illustrate the important outcomes of the study, a conceptual model of Resilience Right Decisions for Electronic Commerce and Cybersecurity was created (Figure 8).

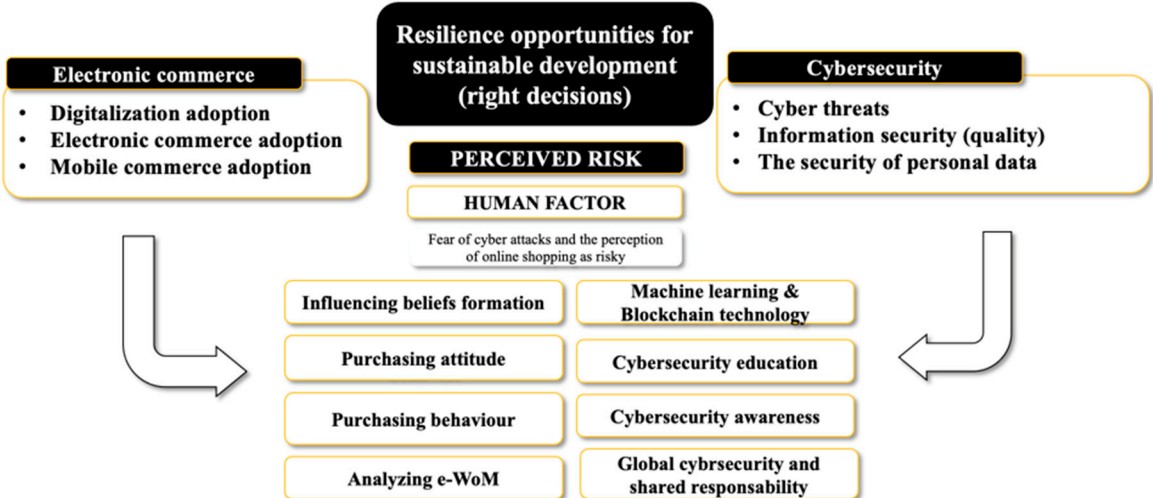

**Figure 8.** The Resilience Right Decisions for Electronic Commerce and Cybersecurity Model.

Electronic commerce, as a vital element in worldwide economic resilience, gains a lot of attention currently from the point of view of implementing the necessary technology to use it. Electronic commerce adoption and mobile commerce adoption are two important challenging research lines for this field in the global economic resilience [27]; the largest group of references by the keywords criteria validated it. Electronic commerce, customer experience, social media, search engine marketing, and big data perspective represent the guideline topics for digital business owners. This fact highlights the global impact.

The study revealed that cybersecurity with specific threats, the quality of information security, and the security of personal data were found as the main challenges for the resilience and the right decisions for stakeholders [26].

The analysis performed centred on the human factor in the middle of the pursuit of the risk faced when cybersecurity is referred to. The fear of cyber-attacks, such as ransomware or other types of cyber threats, is no longer a novel issue. What is now a real concern refers to facing the perception of online shopping as being risky, especially from the point of view of personal data security. Consumers who perceive online shopping as risky and who are concerned about the resilience of the retailer's infrastructure and the security of the personal data [31,32] draw a connection with cybersecurity awareness, which requires the need for fraud mitigation in the online environment. The connection between cybersecurity and electronic commerce adoption asks for cybersecurity education, cyber awareness, data protection with lasting attention to purchasing behaviour, e-WOM, social commerce, and fast shipping to increase the resilience of digital businesses. A relevant outcome of this study revealed that aiding better development requires the use of data, even in the face of challenges related to global cybersecurity. Adding blockchain and machine learning alongside Artificial Intelligence [14] as technologies in use for strengthening resilience can represent the right decision for traders to accelerate the recovery to pre-COVID-19 levels. Additionally, the results of the study can complement other knowledge research fields, such as digital transformation (addressed with bibliometric analysis) [48], cyber threat and cyber-attack literature for higher education [49], but also, a global perspective on cybersecurity trends [50].

## 5. Conclusions

The authors bring to light a summary of the electronic commerce research and cybersecurity knowledge field, strongly connected with global digital resilience, by presenting to the academic world: the big picture, which is easy to understand. The Resilience Right Decisions for Electronic Commerce and Cybersecurity Model promotes the top priorities

and useful information, increasing the awareness of the global impact of cybersecurity on commerce performed in the digital environment.

The managerial implications point to the priority of deploying cybersecurity optimisation actions to shape marketing strategies for purchase, re-purchase behaviour, and technology adoption. The most important detail business owners should understand is related to the nature of the human factor, which influences the competitiveness of organisations and their resilience, especially during hard times. Knowing about factors such as purchase behaviour, customer experience, blockchain technology, big data, and data protection, organisations can develop a strong resilience. Additionally, business owners have the opportunity to deploy solutions regarding the quality and security of the infrastructure and know-how for the safe use of resources: both for customers and employees.

The theoretical implications consist of collaborating with organisations, researchers, and publishers to develop and promote new theoretical frameworks to implement cybersecurity solutions for electronic commerce. Among the types of study they could create in the research field are: sentiment analysis, latent transition analysis, literature reviews, and empirical investigations.

The first limit comes from studying the research papers by only using the Web of Science Core Collection database. The second limit is represented by the probability of including several articles that are less relevant for the analysis because of the selection made automatically by the software. Additionally, the authors decided to choose only two research knowledge fields—the possibilities being endless.

These facts can turn into research directions by extending the number of both research knowledge fields. Additionally, an extension from the Web of Science Core Collection to the SCOPUS database may enlarge the analysis opportunities. It is also possible to conduct direct research on the perception of electronic commerce site owners, regarding the impact of cybersecurity, to develop their businesses.

**Author Contributions:** Conceptualization, D.R.V., E.N., O.M.Ț., A.Z., I.B.C., T.F. and G.B.; methodology, E.N., G.B.; literature review, I.B.C., D.R.V., A.Z.; analysis and writing the results, E.N. and O.M.Ț.; discussion and conclusions, E.N., I.B.C., O.M.Ț., D.R.V., A.Z., T.F. and G.B.; writing—original draft preparation, E.N., I.B.C., A.Z.; writing—review and editing, I.B.C., O.M.Ț., D.R.V., A.Z., T.F. and G.B.; supervision, G.B.; project administration, G.B.; funding acquisition, G.B. All authors have read and agreed to the published version of the manuscript.

**Funding:** This APC was funded by Transilvania University of Brașov.

**Institutional Review Board Statement:** Not applicable.

**Informed Consent Statement:** Not applicable.

**Data Availability Statement:** The data presented in this study are available online at shorturl.at/dfiEP (last accessed on 23 June 2022).

**Conflicts of Interest:** The authors declare no conflict of interest.

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
