# Peer review of "Extending the Frontiers of Electronic Commerce Knowledge through Cybersecurity"

_electronics, doi:10.3390/electronics11142223_

Round 1

Reviewer 1 Report

Please stress the differences between digital resilience and cyber resilience giving the definitio s from literature. The same is necessary for cybersecurity and cyber defence. At page 11 when denoting the figure there should be Figure 8.

Reviewer 2 Report

1. The critical literature review was conducted at a good level. In the bibliography, the authors presented 42 current publications in this field of research, which were included in the presented article.  The oldest item presented in the bibliography is item no. 34. Molla, A.; Licker, P.S. E-Commerce Adoption in Developing Countries: A Model and Instrument. Inform Manage 2005, 42(6), 877-899. http://dx.doi.org/10.1016/j.im.2004.09.002. However, the position is valid for the research presented by the authors of the paper.

2. The drawings presented in the article by the authors are of good quality and are legible for future researchers related to the topic of the article. However, I would suggest that the Authors of the article in Fig. 2 entitled Figure 2. "Visualized comparative analysis: (a) Network for electronic commerce field of research; (b) Network for cybersecurity field of research, designed by authors" they have marked or developed a legend for this image on the drawing, in the form of: a set of green points of various sizes is: .....; the set of blue points is: ........., the same applies to fig. 3 b. A drawing for readers would be clearer. The remaining drawings do not raise any substantive reservations.

3. The authors very well prepare tables with research results - they are clear and contain the most important information relevant to the presented article.

4. The conclusions are short but sufficient - they contain the most important information relevant to the presented article.

5. In the text under the formula No. 1, please add the legend what they mean: k - .....; ci - .......; e.t.c.

6. I would like to ask you to add a table of the most important abbreviations used in the article, before the bibliography. This will make reading the entire article much easier.

Reviewer 3 Report

-This paper presents a knowledge-based model in the form of two networks showing that the human factor is central with regards of the fear of cyber-attacks and the perception that online shopping is risky.

-The technical aspects of the research presented seem sound and detailed.

-I think it is interesting and relevant the model you propose (The Resilience Right Decisions for Electronic Commerce and Cybersecurity Model), as it could support other researchers as well as policy decision makers.

-Are you considering using a larger number of both research knowledge fields as part of your future work?

-Regarding the writing style and typos in the paper, I suggest that you double check all the document as there are some typos.

-The results obtained seem promising and it would be interesting to see results related to other research knowledge fields.

-The network visualisations could be a bit more clearer in terms of the quality of the image and in terms of their content.
